# Orthotopic Bone Formation by Streamlined Engineering and Devitalization of Human Hypertrophic Cartilage

**DOI:** 10.3390/ijms21197233

**Published:** 2020-09-30

**Authors:** Sébastien Pigeot, Paul Emile Bourgine, Jaquiery Claude, Celeste Scotti, Adam Papadimitropoulos, Atanas Todorov, Christian Epple, Giuseppe M. Peretti, Ivan Martin

**Affiliations:** 1Department of Biomedicine, University Hospital Basel, University of Basel, 4031 Basel, Switzerland; sebastien.pigeot@usb.ch (S.P.); paul.bourgine@med.lu.se (P.E.B.); adam@benefis-medical.it (A.P.); atanastodo@gmail.com (A.T.); 2Department of Surgery, University Hospital Basel, University of Basel, 4031 Basel, Switzerland; claude.jaquiery@usb.ch (J.C.); christian.epple.ce@gmail.com (C.E.); 3Novartis Institutes for Biomedical Research, 4056 Basel, Switzerland; celeste.scotti@novartis.com; 4IRCCS Istituto Ortopedico Galeazzi, 20161 Milan, Italy; gperetti@iol.it; 5Department of Biomedical Sciences for Health, University of Milan, 20133 Milan, Italy

**Keywords:** apoptosis, bioreactors, bone repair, endochondral ossification, hypertrophic cartilage, regenerative medicine

## Abstract

Most bones of the human body form and heal through endochondral ossification, whereby hypertrophic cartilage (HyC) is formed and subsequently remodeled into bone. We previously demonstrated that HyC can be engineered from human mesenchymal stromal cells (hMSC), and subsequently devitalized by apoptosis induction. The resulting extracellular matrix (ECM) tissue retained osteoinductive properties, leading to ectopic bone formation. In this study, we aimed at engineering and devitalizing upscaled quantities of HyC ECM within a perfusion bioreactor, followed by in vivo assessment in an orthotopic bone repair model. We hypothesized that the devitalized HyC ECM would outperform a clinical product currently used for bone reconstructive surgery. Human MSC were genetically engineered with a gene cassette enabling apoptosis induction upon addition of an adjuvant. Engineered hMSC were seeded, differentiated, and devitalized within a perfusion bioreactor. The resulting HyC ECM was subsequently implanted in a 10-mm rabbit calvarial defect model, with processed human bone (Maxgraft^®^) as control. Human MSC cultured in the perfusion bioreactor generated a homogenous HyC ECM and were efficiently induced towards apoptosis. Following six weeks of in vivo implantation, microcomputed tomography and histological analyses of the defects revealed an increased bone formation in the defects filled with HyC ECM as compared to Maxgraft^®^. This work demonstrates the suitability of engineered devitalized HyC ECM as a bone substitute material, with a performance superior to a state-of-the-art commercial graft. Streamlined generation of the devitalized tissue transplant within a perfusion bioreactor is relevant towards standardized and automated manufacturing of a clinical product.

## 1. Introduction

Bone repair is among the most potent regenerative processes in the human body, typically healing without scar tissue. In some critical cases, however, spontaneous bone healing is not sufficient and surgical intervention using graft substitutes is required [1].

The current gold standard for bone grafting consists in autologous bone transplant. Nevertheless, there are significant drawbacks, such as limited material availability, increased risks of infection and morbidity at the donor site. Alternative strategies rely on materials with limited biological functionality (e.g., xeno-, allo-graft, synthetic scaffolds) or on osteoinductive growth factors delivered at supra-physiological doses, raising safety concerns [2]. Therefore, there is a crucial need for novel bone repair strategies.

A recently proposed approach for bone repair consists in the recapitulation of endochondral ossification (EO), the physiological healing process where bone tissue is the result of a hypertrophic cartilage (HyC) template remodeling [3,4]. Successful recapitulation of EO using adult human bone marrow-derived mesenchymal stromal cells (hMSC), after their differentiation into engineered HyC, has been demonstrated by many groups including ours [5,6,7,8,9,10,11]. To bypass an autologous setting and enable an off-the-shelf solution, the engineered HyC could be devitalized, leading to an extracellular matrix (ECM) embedding the cytokines capable to activate the process of EO [12,13]. To prevent extensive loss of cytokines from the engineered ECM, we previously developed a devitalization technique based on selective induction of cell death by engineering primary hMSC with a caspase inducible apoptotic system [14]. The resulting HyC pellets, efficiently devitalized upon medium supplementation with a clinical grade adjuvant, could induce bone formation when ectopically implanted [14]. 

Towards clinical fruition of the strategy, in this work we aimed at engineering an upscaled HyC-based material within a bioreactor system, with streamlined ECM deposition and devitalization in the same chamber. The bone graft substitute material performance was subsequently assessed in an immunocompetent orthotopic environment. We hypothesized that such devitalized HyC ECM would enable superior bone regeneration in an orthotopic environment as compared to a clinical standard-of-care ECM-based material, namely an allograft derived from processed human bone (Maxgraft^®^).

## 2. Results

### 2.1. Perfusion Culture Allows the Generation of Upscaled Hypertrophic Cartilage

Primary hMSCs were efficiently transduced with the inducible apoptotic cassette and sorted to obtain a homogenous population. Cells were then seeded and cultured statically or under perfusion flow within collagen based discs, with the goal of upscaling the volume of HyC (56.5 mm^3^) as compared to the typical one previously generated in 1-mm spheres macromass pellets (0.5 mm^3^) [14], thus representing a 113-fold increase (Figure 1). In vitro cartilage formation was first assessed by measurement of glycosaminoglycans (GAG), a major cartilaginous tissue component, both in the supernatant and in the generated tissues. GAG amounts released in the supernatant were 6–20- fold higher in perfusion than static cultures (Figure 2A), whereas those accumulated within the tissue were comparable and at levels typical for effective cartilaginous tissue formation [7,9] (Figure 2B). Gene expression analysis indicated that cells in the static and perfused tissues acquired similar chondrogenic (ColII) and hypertrophic (MMP13, IHH, ColX, BSP) phenotypes (Figure 2C), as compared to undifferentiated hMSCs. The formation of hypertrophic cartilage was histologically confirmed by Saf-O and collagen type X positive staining in both experimental groups (Figure 2D). However, while positive stainings were restricted to peripheral regions in statically cultured tissues, the specimens cultured under perfusion displayed a homogenous pattern reaching the central core. HyC had identical mineralization (Alizarin red) patterns in the outer parts of the tissues, independently of the culture condition (Figure 2D).

Overall, perfusion within the bioreactor chamber allowed for the generation of an upscaled HyC, displaying a similar differentiation stage and quality as statically cultured HyC, but with a more homogenous tissue distribution. Since this feature is essential to reliably generate minced HyC with reduced variability in quality, only perfusion cultured HyC were considered in the subsequent experiments.

### 2.2. Engineered HyC ECM is Efficiently Devitalized under Perfusion Flow

Following validation of the perfusion bioreactor-based generation of HyC, we assessed the efficiency of devitalization by apoptosis induction through the addition of an adjuvant to the medium in perfusion (apoptized), with vehicle supplementation (non-apoptized) as control. As expected, control HyC contained a minimal percentage of dead cells (3.75%), whereas in apoptized HyC, more than 95% of the cells were dead (Figure 2E). The perfusion bioreactor system thus supported an efficient built-in devitalization of upscaled HyC ECM.

### 2.3. HyC ECM Enhances Mineralized Tissue Formation in Calvarial Defects 

The performance of the devitalized HyC ECM graft generated in the bioreactor, in comparison with a commercially available clinical standard-of-care bone substitute (Maxgraft^®^), was tested in an immunocompetent orthotopic rabbit calvarial model. Prior to implantation, the devitalized HyC ECM discs were minced to obtain granules of similar shape and size (≤2 mm) as the Maxgraft^®^ material. In each rabbit, two 10-mm diameter calvarial defects were generated and filled with a volume of 169.5 mm^3^ of either minced HyC ECM (corresponding to three engineered HyC constructs) or Maxgraft^®^ granules. 

After six weeks of in vivo implantation, calvaria were retrieved and mineralization assessed by µCT. The mineralization volumes of each defect were quantified using two different cylindrical regions of interest (Figure 3A). A 10-mm diameter cylindrical region, corresponding to the full defect, was first considered to assess the total newly formed mineralized tissue. For this purpose, the volume of an amount of Maxgraft^®^ corresponding to the one introduced into each defect was measured by µCT (Appendix A) and subtracted from the total mineralized volume, assuming no relevant resorption over the time of implantation. HyC ECM treated defects showed a 15-fold increase in mineralized volume as compared to the Maxgraft^®^ ones (Figure 3B). To reduce influence of the typically vigorous osteoconduction process from the surrounding cortical bone, a 5-mm diameter central cylindrical region was then analyzed. In this volume, HyC ECM treated defects displayed a 3.7-fold higher amount of mineralized tissue as compared to those filled with Maxgraft^®^ (Figure 3C). Thus, HyC ECM were shown to induce stronger mineralization than the Maxgraft^®^ throughout the defect. 

### 2.4. HyC ECM Enhances Bone Tissue Formation in Calvarial Defects

To more specifically assess HyC matrix remodeling and to quantify bone formation, histological sections were processed using typical stainings for bone and cartilage (Figure 4A,B). Negative Safranin-O staining for glycosaminoglycans in the implanted HyC ECM six weeks post-in vivo implantation (Figure 4A,B) indicated a substantial remodeling of the tissue. Positive (brown) staining for Masson allowed to differentiate the mature mineralized areas from those of de novo bone formation, which could then be precisely quantified. The area of newly formed bone tissue was measured at four different levels, with four tissue sections assessed at each level using the H&E staining in bright field and auto-fluorescence to account for the allograft bone auto-fluorescence (Figure 4A,B). The area of bone regions disconnected from the surrounding cortical bone was 2.3-fold higher in the defects filled with HyC ECM than in those filled with the Maxgraft^®^ material (Figure 4C), consistent with mineralization data obtained by µCT in the central cylinders. The area of bone regions formed in direct contact with the surface of the materials was also significantly higher (2.9-fold) in the defects filled with HyC ECM than in those filled with the Maxgraft^®^ material (Figure 4D). To confirm the mouse origin of the formed bone, a staining for human repeated Alu sequences was carried out showing the absence of human cells throughout the defect and particularly in contact with the newly formed bone (Appendix A). Together, these findings indicate that devitalized HyC ECM more efficiently support bone formation as compared to a typical allograft, not only by enhanced osteoconduction, but possibly through a mechanism of osteoinduction.

## 3. Discussion

Our work demonstrates the feasibility to engineer and subsequently devitalize upscaled amounts of HyC ECM in a streamlined process within a perfusion bioreactor system. The resulting engineered but cell-free material, when tested at an orthotopic site, was demonstrated to lead to a superior formation of de novo bone tissue as compared to a clinical standard-of-care (Maxgraft^®^). 

We previously reported that static seeding and culture of hMSC on the same collagen sponge used here as scaffold resulted in upscaled HyC ECM, but with a large necrotic core [15]. Here, we established that the introduction of a perfusion-based bioreactor system strongly enhanced the uniformity of tissue formation, likely due to increased transfer of nutrients/oxygen throughout the tissue [16,17,18], without compromising the process of chondrogenesis and hypertrophy induction. Moreover, the bioreactor system allowed to perfuse the apoptosis-activating compound through the engineered constructs within the same culture chamber, thus fulfilling the vision of a streamlined process combining tissue culture with its devitalization [19,20]. The bioreactor used in this study could be further developed to generate larger amounts of HyC ECM, to incorporate an automated medium exchange as well as to include monitoring/control of oxygen levels and pH, towards standardized and optimized manufacturing of HyC ECM [21,22,23,24]. Additionally, measuring GAG released in the supernatant could provide a reliable and non-invasive tool to monitor tissue development prior to implantation. 

The engineered and devitalized HyC ECM was then tested in an immunocompetent orthotopic defect site. Although the immunogenicity of the devitalized HyC ECM was not extensively assessed, no obvious reaction could be observed. We may speculate that the apoptosis induction, generating apoptotic bodies instead of uncontrolled organelles release through necrosis, may lead to attenuated immune response [25]. Additionally, perfusion of the tissue during the apoptotic process likely removed some of the potential immunogenic material from the tissue. The tissue filling the calvarial defects was analyzed by µCT for mineral content, as well as by quantitative histomorphology for a more stringent assessment of de novo formation of bone, both isolated from the calvarial bone and in contact with the implanted material. The superiority of HyC ECM over the reference Maxgraft^®^ material in the formation of new bone tissue is in line with the superior performance of biological grafts as compared to allografts with a merely inorganic composition [2]. This outcome is most likely due to the preserved proteins embedded in the devitalized HyC ECM, such as inflammatory (e.g., interleukins [4]), angiogenic (e.g., vascular endothelial growth factor (VEGF) [26]), and osteogenic (e.g., bone morphogenetic protein 2 (BMP2) [27]) cytokines. 

In our work, the generation of devitalized HyC ECM requires the use of primary hMSC, which is expected to lead to high variability in the quality of the resulting material depending on the donor [28,29]. Additionally, there is currently no reliable marker allowing to predict the potential of hMSC preparations to form HyC [30]. To overcome this limitation, we envision the engineering of cell lines, capable to reproducibly and efficiently form HyC ECM and at the same time inducible to apoptosis. Such cell lines could also be genetically engineered to overexpress defined factors, enabling controlled customization of the resulting ECMs, enriched in defined factors [31,32,33]. For example, the efficiency of bone formation in specific critical settings could be increased by engineering cells overexpressing cytokines known to enhance the rate of HyC remodeling, the ingrowth of vasculature or the polarization of inflammatory/immune cells [4,34,35].

Standardized generation of HyC ECM as a bone graft material also requires a suitable storage strategy. Major allograft storage processes include cryopreservation, deep freezing (fresh-freezing) and lyophilization (freeze-drying) [36]. Cryopreservation and deep freezing are the most common storage procedures. Despite a recent report suggesting storage at –20 °C [37], frozen tissues are usually kept at –70 °C, leading to logistic issues of shipping and storage [38]. Future research should aim at characterizing the quality and function of stored engineered HyC ECM as a potential graft for bone repair.

In conclusion, we showed that large amount of apoptosis-devitalized HyC ECM can be engineered using bioreactor systems and has the potential to outperform a common clinical standard-of-care (Maxgraft^®^) in an immunocompetent orthotopic environment. Ultimately, by exploiting a developmentally inspired approach to the formation of bone tissue through remodeling of HyC, and by combining perfusion-based culture technologies with a built-in devitalization strategy, we envision the standardized manufacturing of a new class of materials for enhanced bone repair. Such grafts are biologically active through cell-produced molecules, but cell-free and available as off-the-shelf solutions.

## 4. Materials and Methods

### 4.1. hMSCs Isolation and Culture

hMSCs were isolated from human bone marrow aspirates obtained from routine orthopedic surgical procedures involving exposure of the iliac crest, after ethical approval (Ethikkommission beider Basel, Ref.78/07) and informed donor consent. Briefly, marrow aspirate (20 mL volume) was harvested from healthy young donor using a bone marrow biopsy needle inserted through the cortical bone and immediately transferred into plastic tubes containing 15,000 IU heparin. After diluting the marrow aspirate with phosphate buffered saline (PBS) at a ratio of 1:4, nucleated cells were counted and seeded at a density of 3.10^6^ cells/cm^2^ in complete medium supplemented with 5 ng/mL of fibroblast growth factor-2 (FGF-2, R&D Systems) and cultured in a humidified 37 °C/5% CO_2_ incubator. hMSCs were selected on the basis of adhesion and proliferation on the plastic substrate one week after seeding and cultured with complete medium, consisting of α-minimum essential Medium (αMEM) with 10% fetal bovine serum, 1% HEPES (1 M), 1% Sodium pyruvate (100 mM) and 1% of Penicillin–Streptomycin–Glutamine (100×) solution (all from Gibco). The cytofluorimetric profile of the expanded cells corresponds to what was previously reported [7].

### 4.2. Lentivirus Production and hMSCs Transduction

The lentiviral vector carrying the inducible apoptotic system (piCas9) consists of a bidirectional promoter driving the transcription of the inducible modified caspase 9 (iCas9) and of the truncated nerve growth factor receptor (∆NGFR) surface marker [39]. The lentivirus was produced by co-transfecting the Lenti-X™ 293T cell Line (Clontech, Mountain View, CA, USA, cat# 632180) with the piCas9 and the packaging plasmids pCMV-VSVG, pRSV-REV, pRRE (Addgene, Watertown, MA, USA, Plasmid #8454, Plasmid #12253, Plasmid #12251 respectively), using FugeneHD according to the manufacturer’s instructions (Promega, cat# E2311). Over the following days, virus-containing supernatant was collected, passed through 0.45-μm filters, and concentrated by ultra-centrifugation.

Transduction of primary hMSCs was carried out at a multiplicity of infection (MOI) of 1. hMSCs were subsequently expanded up to passage 3, stained for the ∆NGFR antibody (Biolegend, San Diego, CA, USA, 345108) and sorted by flow cytometry (FACS BD SORPAria III sorter) to obtain a purified hMSCs population (≥95%) homogeneously expressing the iCas9 device (hMSCs-iCas9).

### 4.3. Generation of Hypertrophic Cartilage Tissue

Purified hMSCs-iCas9 were seeded and cultured on type I collagen meshes to generate HyC tissues. Scaffolds were cultured either statically in Petri dishes or dynamically in a perfusion bioreactor, as described below. For static seeding, 2 million cells in a volume of 35 µl were manually seeded on collagen scaffolds of 6-mm-diameter and 2-mm-thickness (Ultrafoam, BD, Warwick, RI, USA), deposited in standard 12 well culture plate. After a 1 h incubation time at 37 °C to allow cell attachment, constructs were primed toward chondrogenic differentiation for 5 weeks to achieve HyC tissue formation. Chondrogenic medium, consisting of DMEM supplemented with penicillin-streptomycin-glutamine (Invitrogen, Carlsbad, CA, USA), HEPES (Invitrogen), sodium pyruvate (Invitrogen), ITS-A (Insulin, Transferrin, Selenium) (Invitrogen), Human Serum Albumin 0.12% (CSL Behring), 0.1 mM ascorbic acid (Sigma-Aldrich, St. Louis, MO, USA), 10^−7^ M dexamethasone (Sigma) and 10 ng/mL TGF-β3 (Novartis, Basel, Switzerland), was supplied for 3 weeks. Hypertrophic medium, consisting of DMEM supplemented with 50 nM thyroxine, 10 mM β-glycerophosphate (Sigma), 10^−6^ M dexamethasone, 0.1 mM ascorbic acid and 50 pg/mL IL-1β (Sigma), was supplied for the subsequent 2 weeks.

For perfusion bioreactor culture, collagen scaffolds of 8-mm diameter and 2-mm thickness were soaked in PBS and placed in the bioreactor chamber on an EFTE nylon mesh (Fluorotex Sefar, 09-590/47). Scaffold was stabilized with a 1-mm thick Teflon ring leading to a 6-mm diameter perfusable disc. Two million cells were resuspended in 8 mL of complete medium and injected within the bioreactor system (U-CUP, Cellec Biotek AG, Basel, Switzerland). Medium was perfused overnight directly through the scaffold at a flow rate of 3 mL/min, thus allowing for a dynamic seeding of the cells. Subsequently, constructs were cultured at a flow rate of 0.3 mL/min [16] for 5 weeks in chondrogenic medium, similarly to their static counterpart (Figure 1).

### 4.4. Constructs Devitalization

Following the 5 weeks in vitro differentiation, medium was changed and complemented with the AP20187 dimerizer at a concentration of 100 nM (ApexBio, Houston, TX, USA), to induce apoptosis overnight (Figure 1). Retrieved devitalized tissues were washed once with PBS before assessment using live/dead staining (ThermoFischer scientific, Waltham, MA, USA, L3224) or in vivo implantation.

Devitalization was then assessed using a live/dead assay (Invitrogen) staining living cells with 1μM Calcein-AM and dead cells with 0.1 mM ethidium homodimer. Briefly, samples were washed in PBS and incubated with the staining solution for 15 min at 37 °C. Samples were washed twice using PBS and analyzed directly using a confocal microscope (Zeiss LSM710, Oberkochen, Germany). Several z-stack in each sample were acquired. Automatic image analysis was carried out using ImageJ software and segmentation of the cells to quantify living cells and dead cells.

### 4.5. Histological Analysis

Samples used for histology were embedded in paraffin and sections of 5 μm thickness prepared using a microtome (Microm, HM430, ThermoFischer Scientific). Safranin-O, Alizarin red, hematoxylin/eosin, Masson tri-chrome, collagen type X and Alu stainings were performed as previously described [7,9]. Histological quantification of bone tissue was carried out based on the autofluorescence signal of bone following H&E staining. To obtain a representative bone volume formation, four different central regions were selected, each separated by 400 µm. Bone areas formed discontinued from the calvaria or in direct contact with the implanted material were then measured and averaged for each region on four tissue section.

### 4.6. Glycosaminoglycans (GAG) Measurements

GAG content in culture supernatants was assessed using the Barbosa method [40]. Briefly, 250 μL of collected supernatant was incubated with 1 mL of DMMB solution (16 mg/l dimethylmethylene blue, 6 mM sodium formate, 200 mM GuHCL, all from Sigma Aldrich, pH 3.0) on a shaker at room temperature for 30 min. After centrifugation, precipitated DMMB-GAG complexes were dissolved in decomplexion solution (4 M GuHCL, 50 mM Na-Acetate, 10% Propan-1-ol, all from Sigma Aldrich, pH 6.8) at 60 °C for 15 min. Absorption was measured at 656 nm and corresponding GAG concentrations were calculated using a standard curve prepared with purified bovine chondroitin sulfate (Sigma Aldrich).

For the measurement of GAG content in cartilage tissue, samples were preliminary digested overnight at 56 °C in 1 mL of proteinase K solution (Sigma Aldrich, P2308), and 100 µL of the resulting digested solution was used for the DMMB-GAG precipitation.

### 4.7. Quantitative Reverse Transcriptase Polymerase Chain Reaction (qRT-PCR)

Total RNA was extracted from engineered tissues resulting from the 5 weeks in vitro culture using the Quick-RNA™ extraction kit (Zymo Research, Irvine, CA, USA, R1055), according to manufacturer’s instructions. Following cDNA synthesis (Invitrogen 18080044 & Promega, C1181), qRT-PCR was performed using assay on demand (Applied Biosystems, Foster City, CA, USA) on the following genes: Glyceraldehyde 3-phosphate dehydrogenase (GAPDH, Hs02758991_g1), Indian Hedge Hog (IHH, Hs01081800_m1), SRY-box 9 (SOX9, Hs00165814_m1), Matrix metalloproteinase 13 (MMP13, Hs00233992_m1), Collagen type II (ColII, Hs00264051_m1), Collagen type X (ColX, Hs00166657_m1), Vascular endothelial growth factor (VEGF, Hs00900055_m1), Bone morphogenetic protein 2 (BMP2, Hs00154192), Osteocalcin (OCN, Hs01587814_g1), and Bone sialoprotein (BSP, Hs00959010_m1).

### 4.8. Orthotopic In Vivo Implantation

All studies were approved by the responsible ethics authorities and by the Swiss Federal Veterinary Office (permit 2783). Analgesia was provided one-hour prior to surgery by subcutaneous injection of Buprenorphin (0.05 mg/kg body weight). Anesthetic induction was performed with Ketamin (25 mg/kg body weight) and Xylazin (2.5 mg/kg body weight). Anesthesia with isofluran was maintained on demand and regulated by pulsoximetry. Oxygen was used as a carrier gas at a ratio of 1:2 (O2:N2 O). Prior to surgery the skin of the head was shaved and disinfected. A midline longitudinal incision was made from the nasofrontal area to the external occipital protuberance along the midsagittal suture. Skin and underlying tissues were reflected bilaterally to expose the calvaria and two symmetrical 10 mm wide partial-thickness bone defects were created in both parietal bones lateral to the midsagittal suture using a trephine bur (ACE Dental Implant System, Brockton, MA, US), under constant saline coolant irrigation. To complete the bone defect, a piezoelectric tool (MECTRON, Flexident AG, Stansstad, Switzerland) was used to penetrate the calvarial bone, thus minimizing the risk to the underlying tissue. Special care was taken to prevent damage to the dura mater and venous sinuses.

The calvarial defects were filled with either Maxgraft^®^ material (Botiss dental, Article number: 30040), consisting of devitalized human bone, or with devitalized HyC constructs (Figure 1) (*n* = 5 replicates per group). A total volume of 157 mm3 of graft material was calculated to fill up the created rabbit calvaria defects (10-mm diameter and 2-mm height). To this end, cartilage tissues retrieved from 3 bioreactors were pulled together and chopped to form heterogenous granules of similar size as the Maxgraft^®^ filling material (≤2 mm). Prior to implantation, the resulting engineered granules or Maxgraft^®^ were embedded in a fibrin gel (TISSEEL, Baxter) to stabilize the granules into the defect.

Finally, the periosteum and the scalp were closed with 5-0 Vicryl^®^ resorbable sutures (Johnson & Johnson, St. Stevens-Woluwe, Belgium). The scalp was closed with single knots of 4-0 Vicryl^®^ on the outside. The outside sutures were removed after 10 days. Analgesia was given every 6 h for the first 48 h by subcutaneous injection of Buprenorphin 0.05 mg/kg body weight and in the drinking water during the night. Meloxicam 0.3 mg/kg was administered orally at the end of the procedure and every 24 h thereafter for the first week. At the end of the experiment (6 weeks), the animals were sacrificed under anesthesia (Ketamine and Xylazine) with an overdose of Pentobarbital (120 mg/kg body weight) intravenously.

### 4.9. Micro-computed Tomography (µCT)

Calvaria were retrieved and fixed overnight in 4% formaldehyde at 4 °C. Microtomography of the explants was performed using a tungsten x-ray source at 70 kV and 260 μA with an aluminium filter of 0.5 mm (Nanotome, GE, USA). Transmission images were acquired for 360° with an incremental step size of 0.25°. Volumes were reconstructed using a modified Feldkamp algorithm (software supplied by manufacturer) at a voxel size of 10 μm. Thresholding, segmentation, and 3 D measurements were performed using the VGStudio Max software. After microtomography, samples were decalcified in 15% EDTA solution (Sigma Aldrich) before histology.

## Figures and Tables

**Figure 1 ijms-21-07233-f001:**
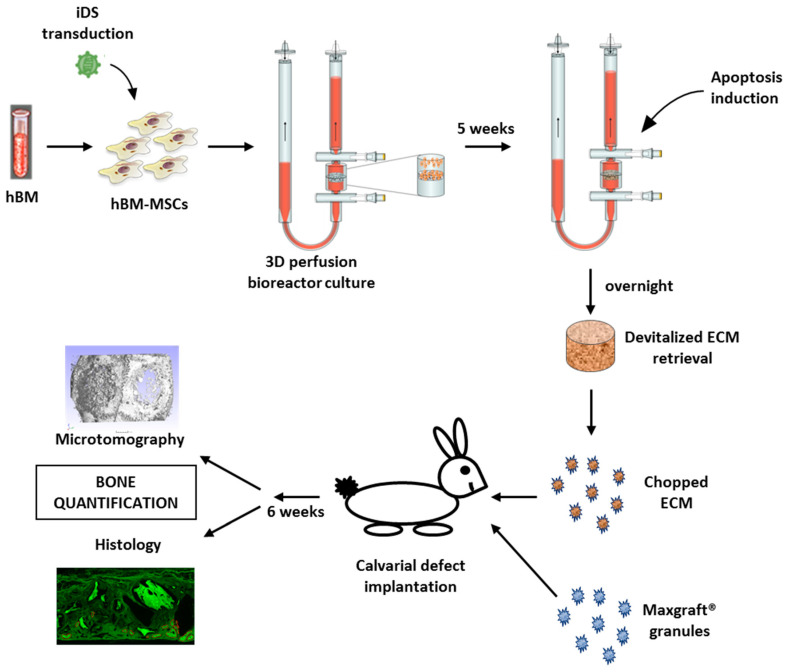
Overview of the experimental plan. Briefly, hMSCs are isolated from total human iliac crest bone marrow (hBM) samples by plastic adhesion. hBM-MSCs are then transduced with lentivirus carrying the inducible caspase 9 (iDS). FACS sorted hBM-MSCs carrying the iDS are then expanded and seeded on collagen sponge within the 3D perfusion bioreactor system. Following the 3 weeks chondrogenic and 2 weeks hypertrophic differentiation protocol, apoptosis is induced overnight in the perfusion bioreactor. HyC ECM are then retrieved, chopped and implanted into 10 mm orthotopic bilateral calvarial defects in combination with the commercially available Maxgraft^®^ granules. Analysis of the calvarial defects is done 6 weeks post-implantation.

**Figure 2 ijms-21-07233-f002:**
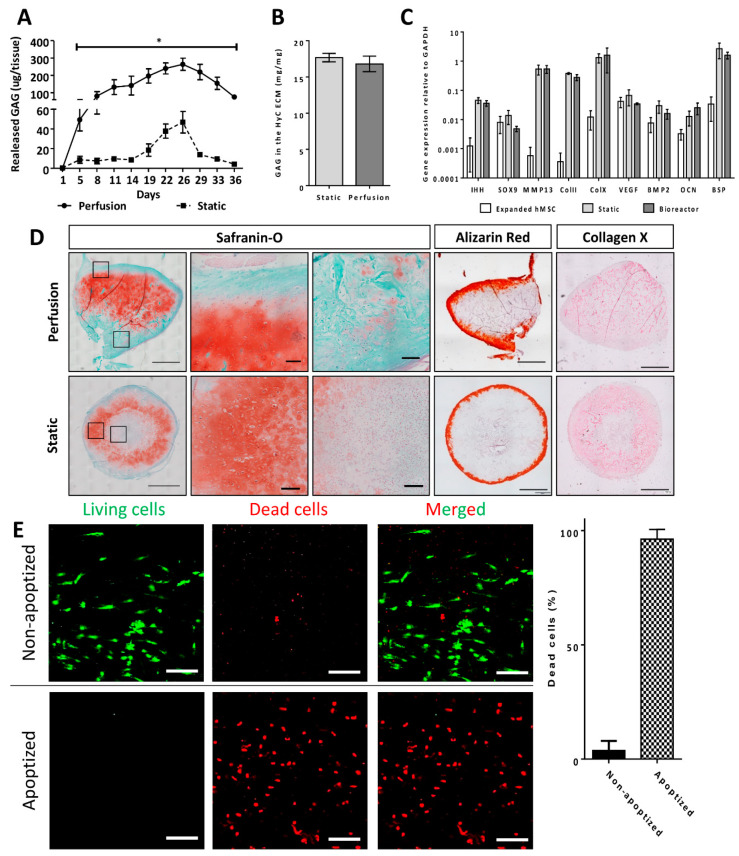
In vitro hypertrophic cartilage (HyC) characterization. (**A**) GAG released quantities in the cell culture media during the 5 weeks in vitro differentiation towards hypertrophic cartilage. (**B**) GAG measured in the HyC ECM after the 5 weeks in vitro differentiation. The GAG is measured per mg of ECM (*n* = 6). (**C**) Gene expression analysis by qRT-PCR following the 5 weeks in vitro differentiation protocol (*n* ≥ 3). (**D**) Sections from the top of the hypertrophic cartilage following 5 weeks in vitro differentiation and before being chopped for orthotopic implantation (scale bar = 1 mm for the whole tissue and 100µm for the zoom-in). GAG is stained in red on the Safranin-O staining (Saf-O). Mineralization stains in red on the Alizarin Red. Collagen type X stains in pink on the immunohistochemistry staining. (**E**) Pictures of the live/dead assay staining in living (Non-apoptized) and devitalized (Apoptized) hypertrophic cartilage and the related cell quantification (scale bar = 200µm) (*n* = 15). Dead cells were represented as a percentage of total quantified cells (living + dead) by imageJ analysis following cells segmentation and quantification. Graphs show the average and SEM. Statistics are two tailed unpaired t-test, * *p* ≤ 0.05.

**Figure 3 ijms-21-07233-f003:**
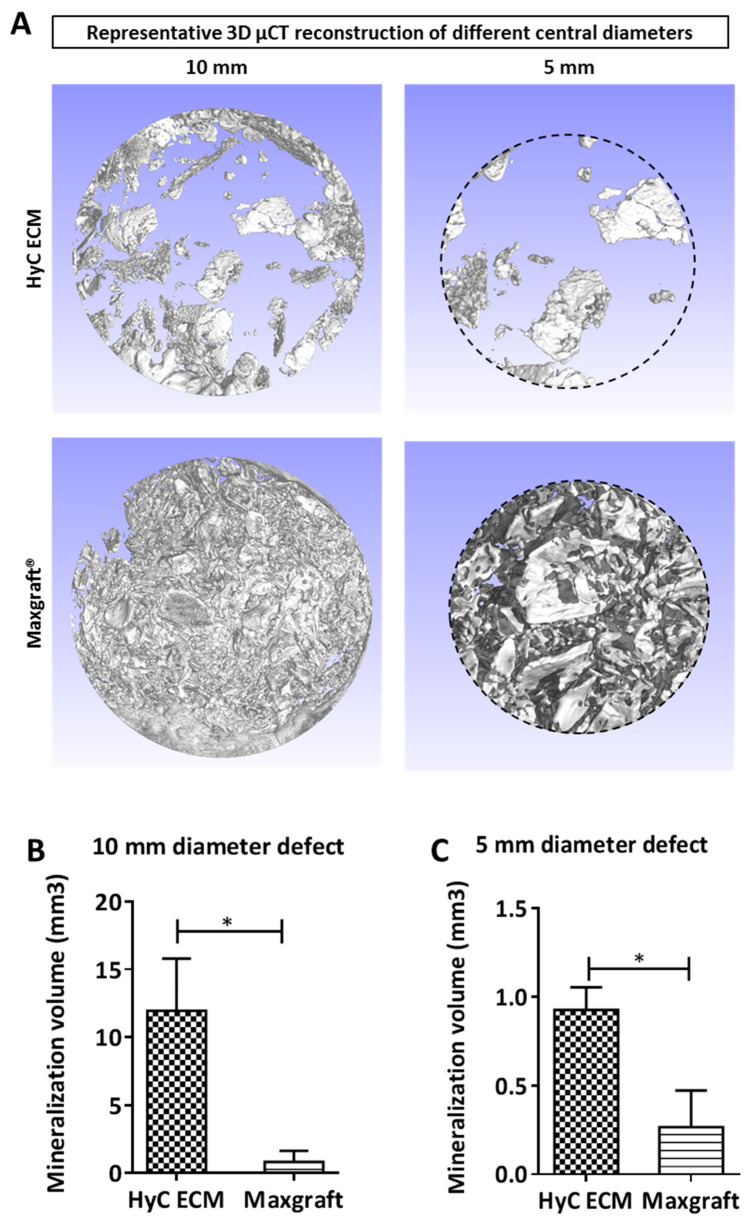
µCT analysis of the calvarial defect. (**A**) 3D µCT reconstruction of the analyzed volumes. (**B**) Mineralization quantification in the 10 mm diameter area and (**C**) in the 5 mm diameter area. Graphs show the average and SEM (*n* = 5). Statistics are two tailed unpaired t-test, * *p* ≤ 0.05.

**Figure 4 ijms-21-07233-f004:**
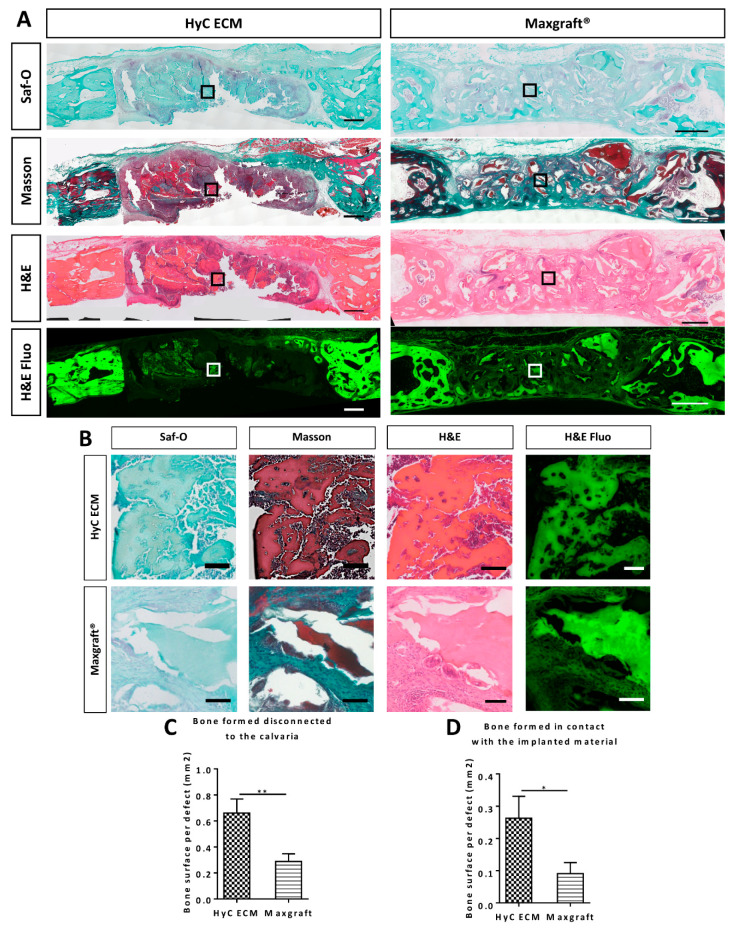
Histological analysis of the retrieved calvarial samples. (**A**) Representative sections of the calvaria 6 weeks after in vivo implantation. Saf-O stains bone in deep green and cartilage in red. Masson stains mature bone in brown and newly formed bone in deep green. H&E stains bone in deep pink. H&E Fluo shows the autofluorescnce of the collagen staining from eosin corresponding to the bone. (scale bar = 1000µm) (**B**) Zoom of squares indicated in part A to highlight bone formation in the calvaria in contact with the implanted material (scale bar = 100µm) (**C,D**) Bone quantified by histomorpho-quantification using H&E staining and tissue autofluorescence. Graph represents the quantified (**C**) total bone disconnected from the calvaria, (**D**) only in contact with the implanted material. Graphs show the average and SEM (*n* = 5). Statistics are two tailed unpaired t-test, * *p* ≤ 0.05 ; ** *p* ≤ 0.01.

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
