# Peer review of "Orthotopic Bone Formation by Streamlined Engineering and Devitalization of Human Hypertrophic Cartilage"

_ijms, 2020, doi:10.3390/ijms21197233_

Round 1

Reviewer 1 Report

Authors report very interesting bone regenertion study with HyC ECM which is superior to Maxigraft. I believe this is very important study for IJMS and I don't have too much issue to publish this manuscript. But I have a few comments to clarify contents before acceptance.

Apoptosis analysis method was missing in the materials and methods section. Please describe in detail.

Figure 2E needs stronger red color intensity. Merged image is also difficult to see. I recommend authors should crop images and enlarge them for better visualization.

I am not sure the definition of "apoptized." I see authors used this term in the previous work. Is it live cells with apoptotic-nuclei? Or just cells undergo apoptosis? If they are still alive, Figure 2E graph shows "dead cells (%)" It means all cells are dead and some showed apoptized character (necrotic or autophagy) and majority with apototic character. Is this what author want to show? Please clarify these in the materials and methods.

Author Response

Response to Reviewer 1 Comments

Authors report very interesting bone regenertion study with HyC ECM which is superior to Maxigraft. I believe this is very important study for IJMS and I don't have too much issue to publish this manuscript. But I have a few comments to clarify contents before acceptance.

Apoptosis analysis method was missing in the materials and methods section. Please describe in detail.

Response: We thank the reviewer for pointing this out and added a short paragraph in the section 4.4 of the Material and methods, line 283 to 287.

Figure 2E needs stronger red color intensity. Merged image is also difficult to see. I recommend authors should crop images and enlarge them for better visualization.

Response: We thank the reviewer for the relevant comment to help making the Figure 2E clearer. We increased the magnification and the contrast of the different pictures and implemented it in the Figure 2 in the manuscript.

I am not sure the definition of "apoptized." I see authors used this term in the previous work. Is it live cells with apoptotic-nuclei? Or just cells undergo apoptosis? If they are still alive, Figure 2E graph shows "dead cells (%)" It means all cells are dead and some showed apoptized character (necrotic or autophagy) and majority with apototic character. Is this what author want to show? Please clarify these in the materials and methods.

Response: We defined as “apoptized” the tissues that were submitted to the apoptosis inducing agent and in which the cells underwent apoptosis. These tissues are also referred to as devitalized. We realize that there was a mistake in the legend of Figure 2 and the terms were inverted. This has now been corrected and clarification introduced in the legend of Figure 2.
Also within the living tissues (Non-apoptized), some cells are expected to undergo apoptosis, due to the natural evolution of a hypertrophic state in cartilage tissue.

Reviewer 2 Report

The manuscript entitled “Orthotopic bone formation by streamlined engineering and devitalization of human hypertrophic cartilage” done by Pigeot et al., engineered orthotopic bone repair model Engineered hMSC demonstrates the suitability of engineered devitalized HyC ECM as a bone substitute material, with a performance superior to state-of-the-art commercial graft models are highly recommended in orthopedic bone repair surgeries and commercial graft development.

The work is oriented towards the characterization and validations of engineering and devitalizing upscaled quantities of HyC ECM within a perfusion bioreactor, followed by in vivo assessment in an orthotopic bone repair model data results are showing high-quality science.

The methodology followed was really impressive and interesting. The authors have well utilized the perfusion culture that allows the generation of upscaled hypertrophic cartilage and validations in vivo validation of the perfusion bioreactor-based generation to engineered HyC ECM is more impressive and valuable to de novo bone tissue graft developments. The methods were explained in a detailed and systematic manner and the corresponding results were discussed in an interactive way. The paper has been written well with clear conclusions. Beyond the potentiality of the manuscript, I have no doubt this paper is great impressive in the field of bone repair clinical surgery researchers and grafts developers of bone tissue remodeling grafts.

Author Response

We thank the reviewer for the very nice comments.

Reviewer 3 Report

It was my pleasure to review the article written by Pigeot et al. proposing an innovative strategy for bone repair.

The article is clear and well written. 

I would like to discuss with the authors two items:

1) Why hMSC were not characterized by flow cytometry or at least by trilineage differentiation?

2) Concerning the histology study. Do you have better images of the Masson trichrome staining (Figure 4) showing the new bone formation in HyCECM that sustains your description of superior new bone formation in your results section? (The resulting engineered but cell-free material, when tested at an orthotopic site, was demonstrated to lead to a  superior formation of de novo bone tissue as compared to a clinical standard-of-care (Maxgraft®).

Author Response

Response to Reviewer 3 Comments

It was my pleasure to review the article written by Pigeot et al. proposing an innovative strategy for bone repair.

The article is clear and well written. 

I would like to discuss with the authors two items:

  • Why hMSC were not characterized by flow cytometry or at least by trilineage differentiation?

Response: The protocol used for hMSC selection and expansion (i.e., bone marrow mononuclear cell plating and passaging) is a standard one used in our group and by many others. We have now modified the text (lines 247-248) to indicate that the cytofluorimetric profile of the resulting cells corresponds to the one described in previous publications (see for example Scotti et al. PNAS, 2010). With regard to trilineage differentiation, this is indeed variable in various preparations from different donors. However, since our approach requires only cartilage formation as a basis to recapitulate endochondral bone formation, we focused only on the chondrogenic differentiation of the expanded cells.

  • Concerning the histology study. Do you have better images of the Masson trichrome staining (Figure 4) showing the new bone formation in HyCECM that sustains your description of superior new bone formation in your results section? (The resulting engineered but cell-free material, when tested at an orthotopic site, was demonstrated to lead to a  superior formation of de novo bone tissue as compared to a clinical standard-of-care (Maxgraft®).

Response: We thank the reviewer for this comment and replaced the image to better show the new bone formation on the Masson trichrome staining.